# The Influence of Ni on Bainite/Martensite Transformation and Mechanical Properties of Deposited Metals Obtained from Metal-Cored Wire

**Jiamei Wang** [1,2], **Xinjie Di** [1,2], **Chengning Li** [1,2,*] and **Dongpo Wang** [1,2,3]

1 School of Materials Science and Engineering, Tianjin University, Tianjin 300350, China; wangjiamei@tju.edu.cn (J.W.); dixinjie@tju.edu.cn (X.D.); wangdp@tju.edu.cn (D.W.)
2 Tianjin Key Laboratory of Advanced Joining Technology, Tianjin 300350, China
3 State Key Laboratory of Metal Material for Marine Equipment and Application, Anshan 114009, China
* Correspondence: licn@tju.edu.cn; Tel.: +86-022-27405889

**Abstract:** The multi-pass deposited metals were prepared by metal-cored wire with low (2.5 wt%) and high (4.0 wt%) Ni to research the effect of Ni on the bainite/martensite transformation. Results showed that deposited metals exhibited a multiphase structure comprised of bainite, martensite and residual austenite, which is not only explained from SEM/TEM, but also identified and quantified each phase from crystallographic structure through XRD and EBSD. With Ni content increasing, the fraction of martensite increases from 37% to 41%, and that of bainite decreases from 61% to 55% accordingly because 4% Ni element narrows the temperature range of the bainite transformation ~20 °C. The 7.8% residual austenite exhibited block and sheet in the deposited metal with low Ni, while the fraction of residual austenite was 3.26% as a film with high Ni, caused by different transformation mechanisms of bainite and martensite. The tensile strengths of deposited metals were 1042 ± 10 MPa (2.5% Ni) and 1040 ± 5 MPa (4% Ni), respectively. The yield strength of deposited metals with high Ni was 685 ± 18 MPa, which was higher than low Ni due to the high fraction of martensite. The impact values of deposited metals with high Ni content decreased because the volume fraction of bainite and residual austenite and area fraction of large-angle grain boundary were lower.

**Keywords:** deposited metal; bainite; strength; toughness; nickel content

## 1. Introduction

High strength low-carbon steels with good toughness and welding properties are appearing materials applied in mechanical engineering [1,2]. Usually, the improvement of strength is related to damage in ductility, limiting structural applications of it without the suitable welding material [3,4]. Thus, developing matching welding materials with appropriate microstructure is necessary. One approach is to prepare the microstructure comprised of bainite, martensite and residual austenite (RA) [5,6]. This microstructure was proved to exhibit higher toughness than full martensite because bainite can separate prior to austenite grain and refine the martensite [7]. Meanwhile, the RA is beneficial to the ductility of materials. The deposited metal is prepared by fluxed-cored arc welding (FCAW, welding process: 138), in which flux-cored wires are not only consumable but also used as the electrode. This welding method [8,9] is widely used in particular in shipbuilding, construction and machine industry as well as regeneration of machine elements, technical devices, manufacturing and modification of surface layers, because of higher welding efficiency, penetration depth, better arc stability, strong adaptability and relative ease of welding. Pure metal powder wires were used in this experiment. The welding wire is kept under strict conditions to prevent corrosion and moisture absorption because it is not a seamless wire. Dirt and moisture inside and on the surface of the

flux-cored wire can get into the welding pool and later into all welded joint zones during welding. This can result in a reduction in the performance of such a welded joint or even its complete destruction [10]. In recent years, the metal-cored wire has been of great importance and developed rapidly with the development of high efficiency and good welding operations [11,12]. More importantly, the alloy elements in the metal-cored wire can be adjusted according to the requirements. Thus, adjusting the composition of the alloying elements in the metal-cored wire, such as Mn, Mo, Cr, Cu and Ni, is the primary method used to form multiphase microstructure [13]. Among the alloying elements, the Ni element can both improve strength and toughness. Ni can decrease the bainite/martensite transformation temperature, strengthen via solid solution and refine grain size [14]. Ni is an austenite stabilizing element and it can reduce ductile-to-brittle transition temperature (DBTT), owing to the reduction of the interaction energy between dislocation and interstitial atoms [15,16]. Xue et al. [17] found that the percentage of ferrite and martensite of deposited metals increased significantly with the increase in Ni content, while bainite decreased. Keehan et al. [18] found the existence of coalesced bainite seriously deteriorated the impact toughness by adding 3.14%, 7.23% and 9.23% Ni. Thus, the excessive Ni content will cause the massive martensite and bainite structures, which preferentially provide paths for crack propagation and deteriorate toughness [19,20]. Therefore, the Ni content in the metal-cored wire is important.

The multiphase microstructure correlated with mechanical properties gives rise to numerous studies. Rao et al. [21] found that the mixed microstructure included bainite led to the improvement of toughness without reduction of the strength, compared with the single martensitic. Tomita [22] proved that lower bainite appeared in the form of acicular, dividing the prior austenite grain with martensite, which provided greater mechanical properties. However, the mechanical properties of upper bainite–martensite decrease. Parsa Abbaszadeh [23] thought that the mixed microstructures containing 12–28% bainite showed higher yield strength than single martensitic because there is a plastic constraint effect induced by the surrounding relatively rigid martensite, leading to improved strength of bainite. The research above serves to study the effects of multiphase on the mechanical properties.

The volume fraction of bainite, martensite and RA is important to the mechanical properties of deposited metals. So, it is essential to identify and quantify them. The traditional optical microscopy (OM) based on color-etched is not adequate in the present case. Because the color-etching mainly depends on partial carbon content [24]. Thus, the electron backscatter diffraction (EBSD) based on the crystallographic structure was employed. Image quality (IQ) values were used to identify lattice defects such as grain boundaries, dislocation, or substructures. The IQ value is high when the quality of the Kikuchi bands is good [25]. Martensite and bainite have unlike degrees of the lattice distortion, which can be translated into different IQ values although they are all BCC structures [26]. Compared to bainite, martensite is shown with lower pattern quality. This method was successfully applied in similar materials with the deposited metals [27].

In this study, the deposited metal was obtained by gas metal arc welding (GMAW). The key challenge is identifying and quantitative analyzing the volume fraction of bainite, martensite and RA, ulteriorly obtaining the metal-cored wire with high strength and good toughness. The effect of Ni on phase transformation was systematically studied by Jmatpro software. The volume fractions of bainite and martensite were quantitatively investigated with the EBSD. The relationship between the fraction of bainite, martensite, and RA and the properties of deposited metal was studied comprehensively.

## 2. Materials and Methods

The metal-cored wire in this experiment was designed by authors and made by a certain factory. The diameter of the wire is 1.2 mm and filling rate is about 15%. They are not seamless wires. The dimension of the steel sheath (99.6%) outside is 12 mm × 0.5 mm. The flux inside is metal particles. The formation process of wires included four steps: U-groove pressing formation, flux filling, rolling sealing and diameter reduction, as shown

in Figure 1 [28]. The base metal used in this experiment was Q345 steel, which is used as the substrate with no effect on sampling, playing a supporting role to prevent molten pool flow and improve molding. The yield strength of Q345 is about 345 MPa with an elongation of more than 22%. The chemical compositions of deposited metal measured by Optical Emission Spectrometer DF-100 (Shenzhen Cepu Technology Co., LTD, Shenzhen, China) are introduced in Table 1. As for the content of Ni, it should not be too high considering the perspective of economic cost. It is mentioned by Mao [29] that when the Ni content reaches 6%, thermal cracks will occur in the weld and seriously damage its mechanical properties. On the other hand, the Ni content should not be too low to ensure the beneficial effect of Ni element on impact toughness. Norstrom et al. [30] found that the ductile-brittle transition temperature would decrease by 20 °C with an increase of 1% Ni. Thus, the 2.5 and 4% Ni wires were manufactured. The deposited metals with 2.5 wt% Ni and 4.0 wt% Ni were labeled as LN and HN, respectively. The deposited metals were prepared by FCAW using a mixture of 80% Ar + 20% $CO_2$ shielding gas. The welding machine was Millermatic 350 (Miller Electric Manufacturing Co., Appleton, WI, USA). The welding current was 220–250 A and arc voltage was 28–30 V. The welding speed was maintained at 300 mm/min. The wire extension is about 14 mm. The flow rate of shielding gas is about 20 L/min. The preheating and inter-pass temperature was 150 °C. The wire is dried before welding.

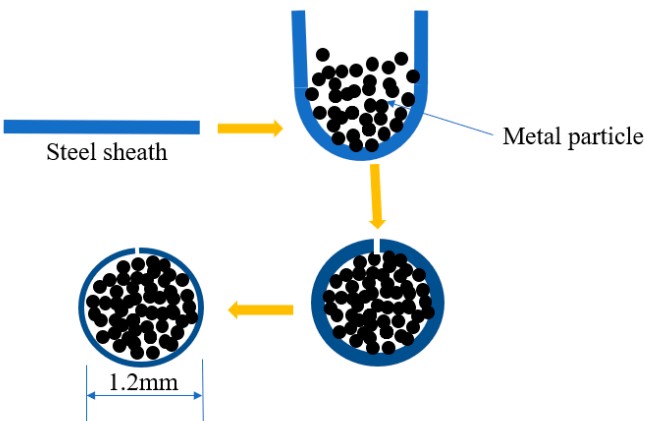

**Figure 1.** Metal-cored wires formation process.

**Table 1.** Chemical compositions of deposited metal (wt%).

|  | C | Mn | Si | Ni | Cr | Mo | S | P | O | N | Ti | Al | Fe |
|---|---|---|---|---|---|---|---|---|---|---|---|---|---|
| LN | 0.080 | 1.55 | 0.50 | 2.43 | 0.84 | 0.72 | 0.0075 | 0.015 | 0.084 | 0.012 | 0.004 | 0.03 | Bal |
| HN | 0.074 | 1.28 | 0.51 | 4.29 | 0.65 | 0.62 | 0.009 | 0.015 | 0.080 | 0.008 | 0.004 | 0.029 | Bal |
| Q345 | ≤0.20 | ≤1.70 | ≤0.50 | 0.50 | 0.30 | 0.10 | 0.035 | 0.035 | – | 0.012 | 0.20 | 0.015 | Bal |

Microstructure investigations perpendicular to the welding direction were measured by scanning electron microscopy JSM-7800F (JEOL, Tokyo, Japan), with an etching solution consisting of 4% nitric acid alcohol. The transmission electron microscopy JEM-2100 (JEOL, Tokyo, Japan) equipped with an ultra-thin-window Oxford energy-dispersive spectrometry (Oxford, Tokyo, Japan). The EDS was used to examine the oxides. For TEM, 3.0 mm diameter disc specimens were wet ground to 50 μm in thickness. The TEM specimens were prepared by twinjet polishing with 95 vol% alcohol and 5 vol% perchloric acid solution. The specimens used in electron backscattered diffraction EDAX-TSL (JEOL, Tokyo, Japan) were electropolished by the 5 vol% perchloric acid alcohol solution (voltage of 30 V). The EBSD data were obtained at an accelerating current of 13 A and a step size of 0.15 μm.

The specimens were also conducted by X-ray diffraction D8 Advanced (Bruker AXS, Bruker, Germany) from 35° to 105° with Cu at the scanning rate of 0.01°/min. The Rietveld

refinement method was applied to calculate the fraction of RA by counting the integrated intensities of (111), (200), (220), (311) austenite peaks and (110), (200), (211), (220) ferrite peaks [29]. The method is expressed as follows:

$$V_\gamma = 1.4I_\gamma / (I_\alpha + 1.4I_\gamma) \tag{1}$$

where $I_\gamma$ is the average integral strength of austenite diffraction peaks, $I_\alpha$ is the average integral strength of ferrite diffraction peaks. The dislocation density was calculated by fullwidth at half-maximum (FWHM) values with a modified Williamson–Hall (MWH) method [31,32]. The MWH method is as follows:

$$\Delta K = \frac{0.9}{D} + \left(\frac{\pi A b^2 \rho}{2}\right)^{1/2} \left(K\overline{C}^{1/2}\right) + \left(\frac{\pi A'^{b^2} Q}{2}\right)^{1/2} \left(K^2\overline{C}\right) \tag{2}$$

where $\Delta K = (2\cos\theta(\Delta\theta))/\lambda$, $K = 2\sin\theta/\lambda$, $\theta$ is the diffraction angle, $\lambda$ is wavelength of X-rays, $D$ is crystallite size, $b$ is burgers vector ($b = 0.2520$ nm) [33], $\rho$ is dislocation density and $\overline{C}$ is average contrast factor of the dislocations, respectively.

The all-weld tensile specimens were machined longitudinally from the weld deposits. Tensile testing was conducted at a strain rate of 0.5 mm/min on an electronic material testing machine MTS Exceed E45 (MTS, Eden Prairie, MN, USA). The impact Charpy V-notch specimens were notched perpendicularly to the welding direction, according to ASTM E23 standard. The specimens were carried out at −20 °C on a JB-300B pendulum with velocity of 5.2 m/s (SANS, Jinan, China). The impact oscilloscope was used to record the crack initiation, propagation and total impact absorbing energy. Three tensile and Charpy V-notch specimens were conducted for each condition. The sampling location and size of specimens are shown in Figure 2. Butt welding was used and the gap between the bottom groove is about 35 mm, ensuring that the influence of substrate on the deposited metal was excluded during the sampling process. The specimens of microstructure were taken from the capping weld bead in as-welded condition. The full-weld tensile/impact sample is taken from the deposited metal between the substrate.

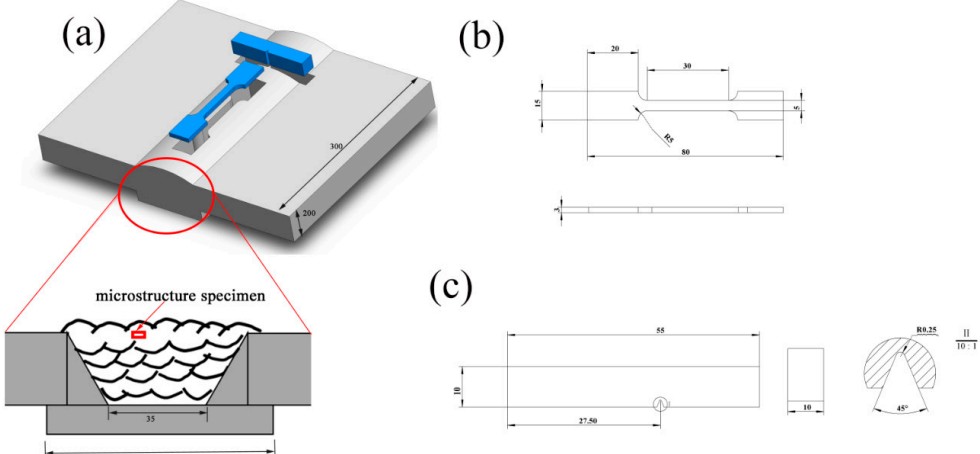

**Figure 2.** Sampling location (**a**) and size of tensile (**b**) and impact specimens (**c**) (mm).

## 3. Results

### 3.1. Microstructural Characterization

Figure 3 shows the SEM images of deposited metals. The specimens of microstructure were taken from the capping weld bead in the as-welded condition. The fine microstructure of deposited metal is a multiphase included bainite, martensite and RA. There are mainly lath bainite and M-A constituents of the LN (Figure 3a). As for the HN, the microstructure is bainite, lath martensite and M-A constituents (Figure 3b).

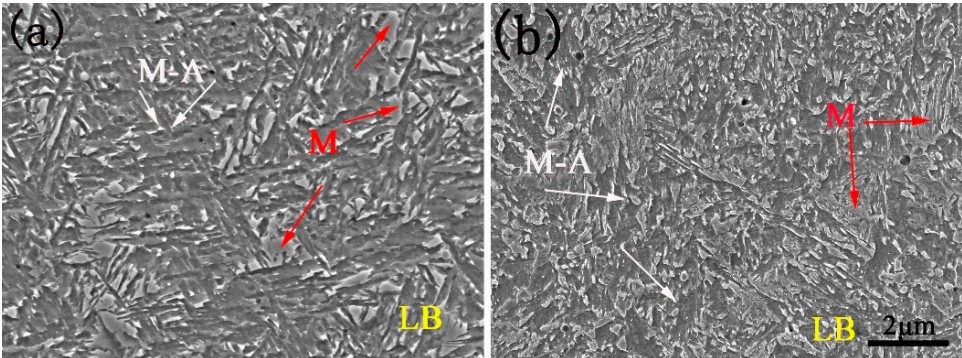

**Figure 3.** SEM images of deposited metals. (**a**) LN, (**b**) HN.

The microstructure of deposited metals is observed by TEM, as exhibited in Figure 4. There is the lathy microstructure of bainite/martensite, and RA distributes between the laths and the tangled dislocation exists inside the laths. The orientation relationship between RA and matrix calculated by selected area electron diffraction (SAED) is approximately consistent with Kurdjumov–Sachs (K-S) orientation [34]: $[110]\gamma//[100]\alpha$ and $(\bar{1}11)\gamma//(011)\alpha$. The bright- and dark-field images of RA are also observed (Figure 5). The RA mainly exists in blocks and sheets of the LN, while it is thin films of the HN. No cementite was observed in the LN or HN.

Figure 6 shows the inverse pole figure (IPF), image quality (IQ) with misorientation angles, and distribution of equivalent grain size. There is the lathy structure with similar crystallographic directions inside one prior austenite grain of the LN and HN. The black parts are oxides that cannot be identified by electron beam resulting in the lowest IQ value (Figure 6b,e). The grain boundaries with 2–5°, 5–15° and 15–180° are represented as red, green and blue lines, respectively. There are mainly small angle grain boundaries inside the bainite/martensite. The grain boundaries between bainite, martensite and RA are large-angle grain boundaries. The percentage of grain boundaries with a large angle (≥15°) of the LN is 61.80%, and that of the HN is 56.23%. Meanwhile, the equivalent grain size is about 4.25 μm of the LN and HN.

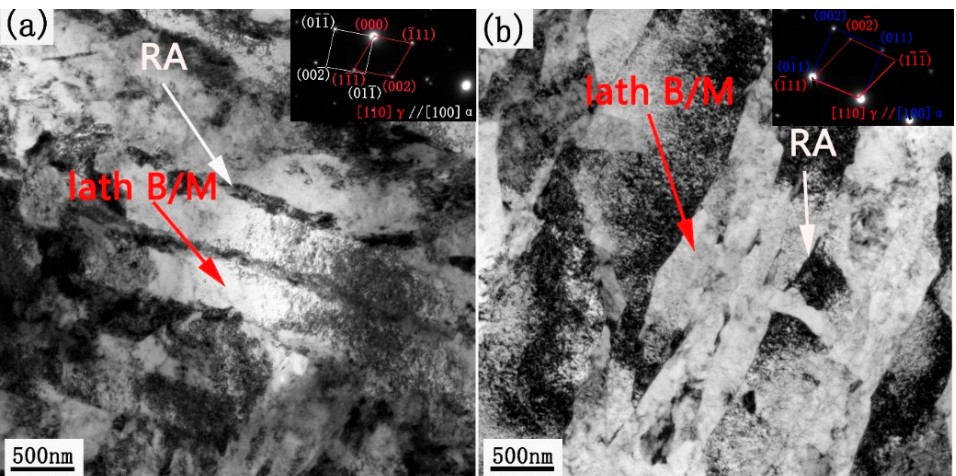

**Figure 4.** TEM images of deposited metals. (**a**) LN, (**b**) HN.

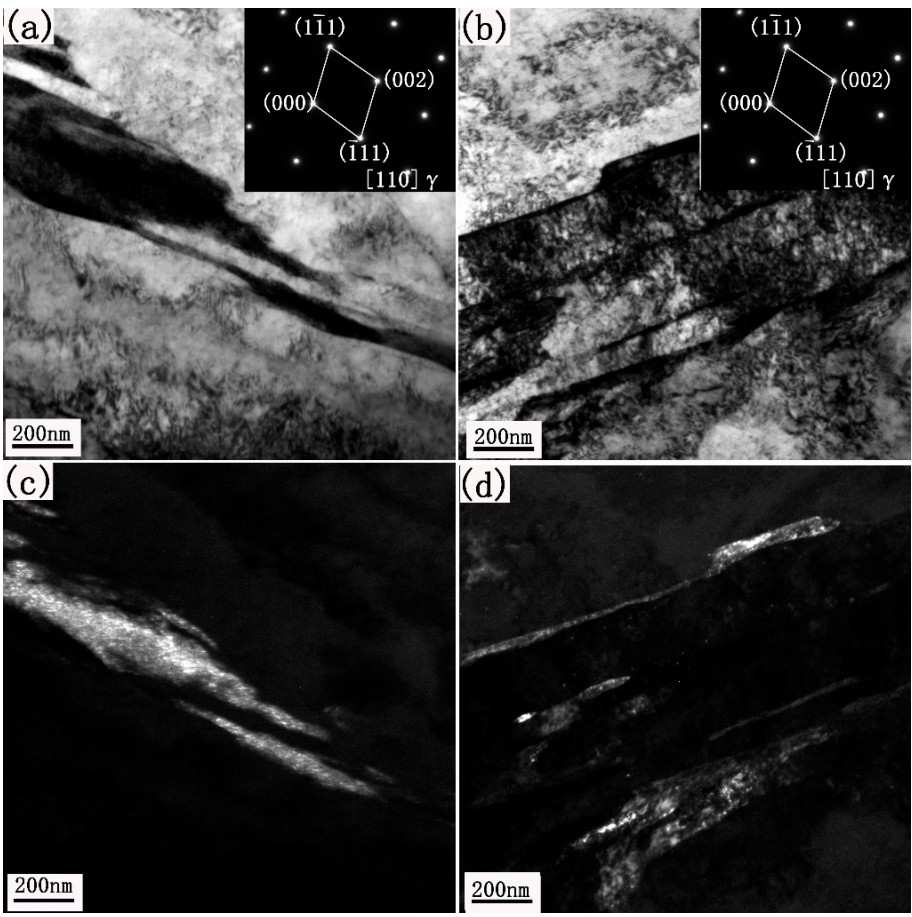

**Figure 5.** Bright-field and dark-field images of RA for deposited metals. (**a**,**c**) LN, (**b**,**d**) HN.

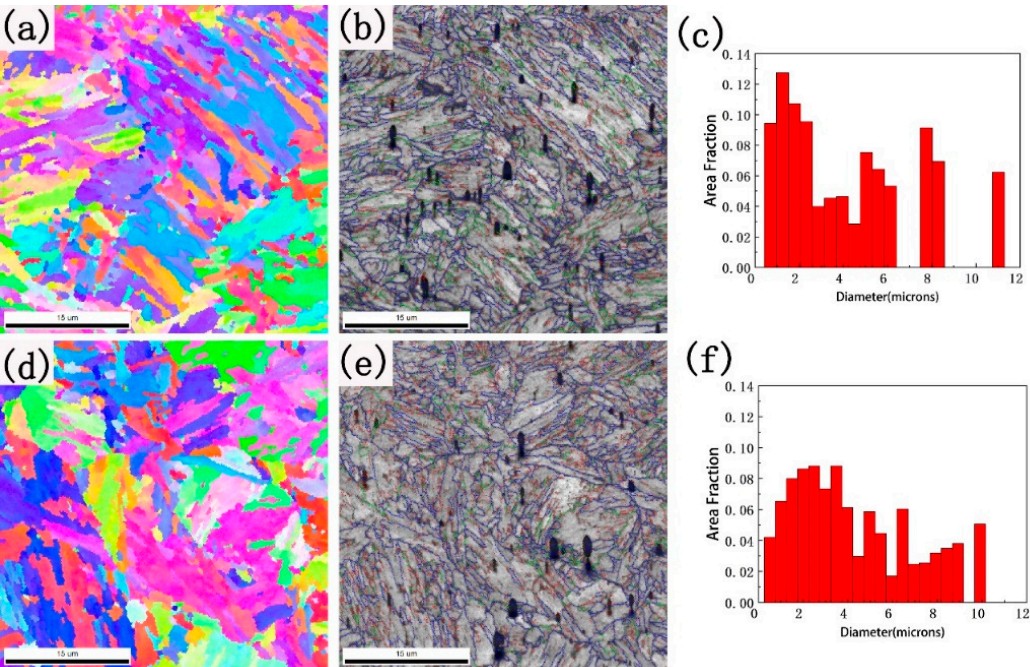

**Figure 6.** EBSD IPF maps (**a**,**d**), IQ maps with misorientation angles (**b**,**e**) and distributions of grain size (**c**,**f**) of deposited metals. (**a**–**c**) LN, (**d**–**f**) HN.

### 3.2. Mechanical Properties

The tensile properties of deposited metals are shown in Figure 7a. The tensile strengths are 1042 ± 10 MPa of the LN and 1040 ± 5 MPa of the HN, and the 0.2% yield strength is about 648 ± 20 MPa and 685 ± 18 MPa, respectively. Meanwhile, the elongation of deposited metals is about 8%. The impact curves are shown in Figure 7b. The impact energy is 46.5 ± 2.8 J at −20 °C of the LN. As for the HN, the value of toughness is 38.8 ± 2.5 J. The total impact energy decreases with the increase in Ni content.

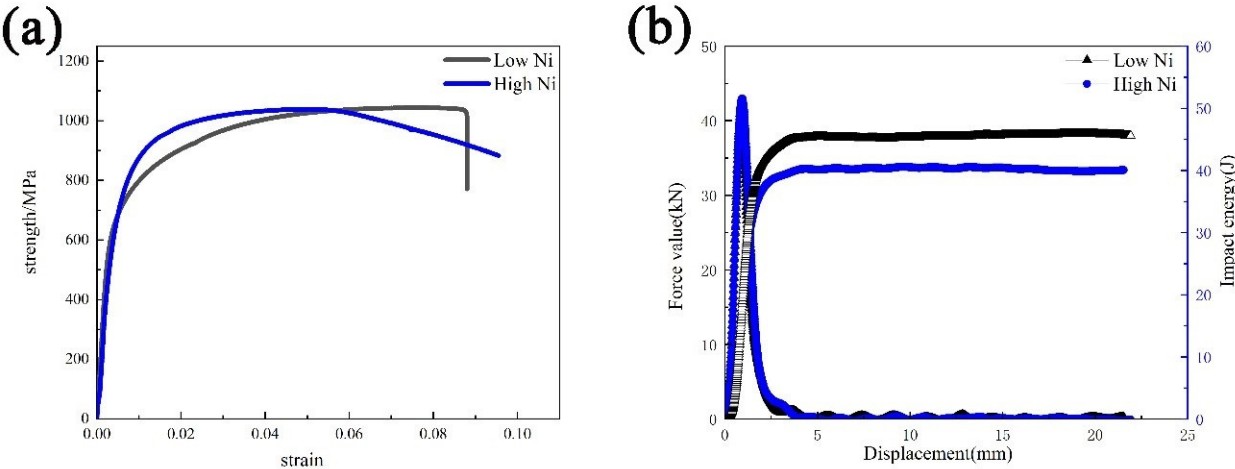

**Figure 7.** Tensile properties (**a**) and impact curves (**b**) of deposited metals.

The impact fracture is observed in Figure 8. The fracture morphology is dimple rupture of the LN and quasi-cleavage of the HN, indicating ductile and quasi-cleavage fracture. With the increase in Ni, the fracture morphology of deposited metals presents a transition from toughness to brittleness. The cleavage surface shows a river pattern, and the cracks start near the circular oxide particles for the HN (Figure 8b). Energy spectrum detection reveals that the composition of oxides is Mn, Si, Al and O (Figure 9). Since oxides are products of voids and microcracks, the toughness is usually reduced by the formation of oxides [35]. Compared with similar materials, the lower toughness value is caused by the high oxygen content of deposited metal, which is about 800 ppm.

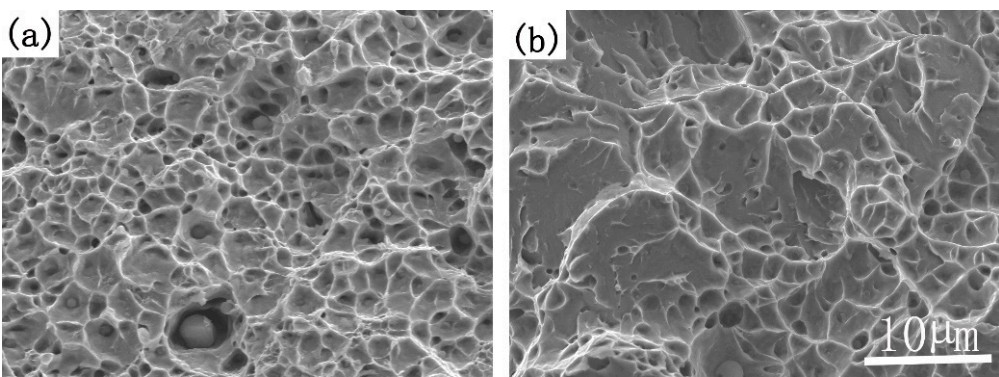

**Figure 8.** SEM of the radioactive area of impact fracture of deposited metals, (**a**) LN, (**b**) HN.

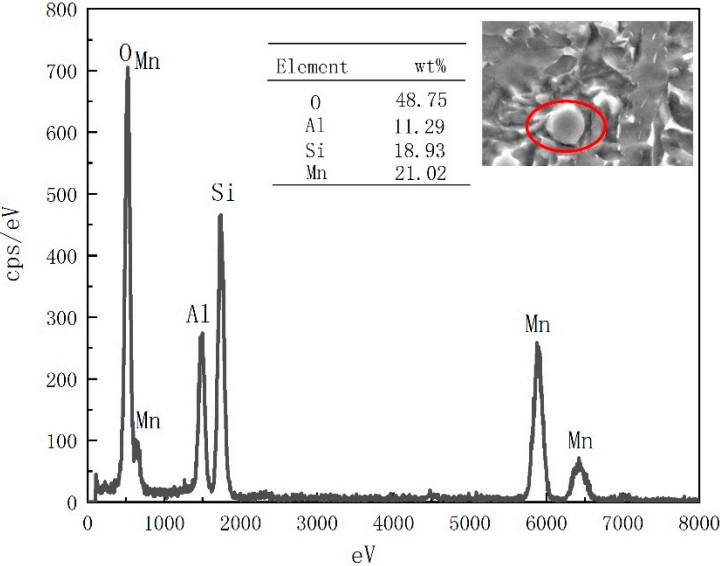

**Figure 9.** SEM micrograph and EDS for oxide of the HN.

## 4. Discussion

### 4.1. Identification of Multiphase and Effect of Ni on Phase Transformation

The multiphase microstructure was analyzed by SEM and TEM before (Figures 3–5), there it was verified by misorientation angles (Figure 10). The misorientation angle between α-Fe bainite/martensite and RA is about 45° [36]. Meanwhile, the 50–60° misorientation angles are grain boundaries of bainite and martensite. The authors [37] have pointed out that the misorientation angle between bainite and prior austenite in the N-W relationship is approximately 53–54°<110> and that between martensite and prior austenite in the K-S relationship is 60°<111>. Thus, the crystallographic morphology of deposited metals consists of bainite, martensite and RA.

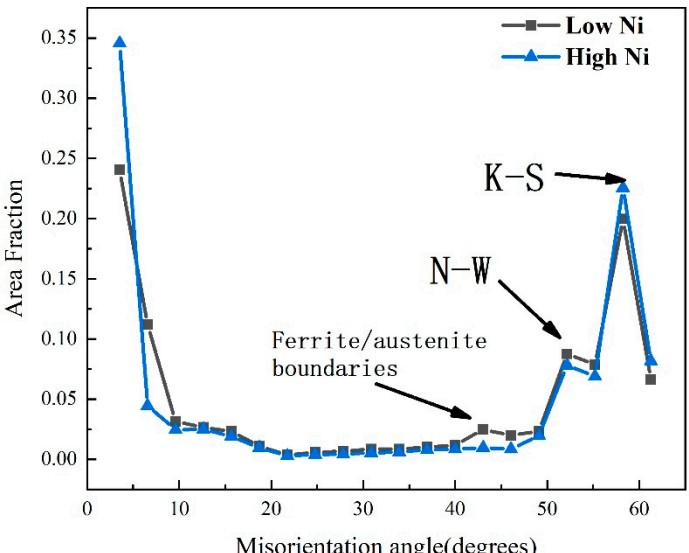

**Figure 10.** Distributions of area fractions of misorientation angles of deposited metals.

The image quality (IQ) values are calculated to quantify the volume fraction of bainite and martensite (Figure 11). The IQ, which is lower than the threshold value determined by a Gaussian fitting, is carried out on the green curve which is segmented as martensite and the red line represents bainite. The smallest peak represents the oxides and null values which cannot be detected by EBSD. The fraction of martensite and bainite can be deduced

by integrating the curves. The results show that the volume fraction of martensite increases from 37% to 41%, while that of bainite decreases from 61% to 55% with the content of Ni increasing.

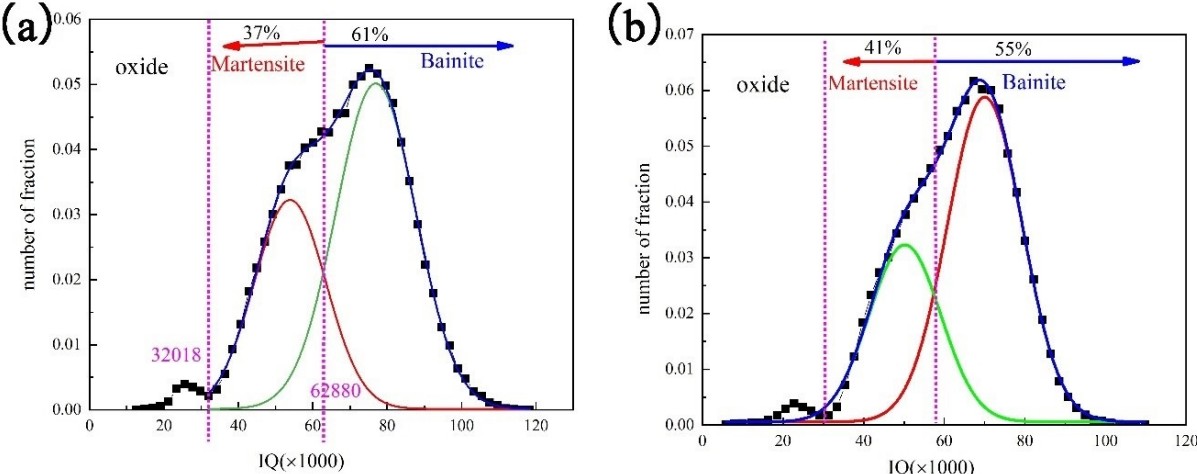

**Figure 11.** IQ chart of deposited metals (**a**) LN, (**b**) HN.

In order to explain the influence of Ni content on bainite/martensite transformation of deposited metal, the continuous cooling transition (CCT) diagram was simulated by Jmatpro software (Figure 12). The cooling rate calculated by the Rosenthal 3D equation [38] at 500 °C during the welding process is about 26.3 °C/s. With the increase in Ni content, the CCT curve of bainite transformation shifts to the lower right. The temperature range of bainite changes from 82.0 °C to 60.2 °C (Table 2). Thus, there is a shorter range of bainite transformation during welding of the HN and the temperature range of martensite almost does not change. Therefore, the high Ni content narrows the temperature range of bainite transformation, lessening the volume fraction of bainite and improving martensite transformation.

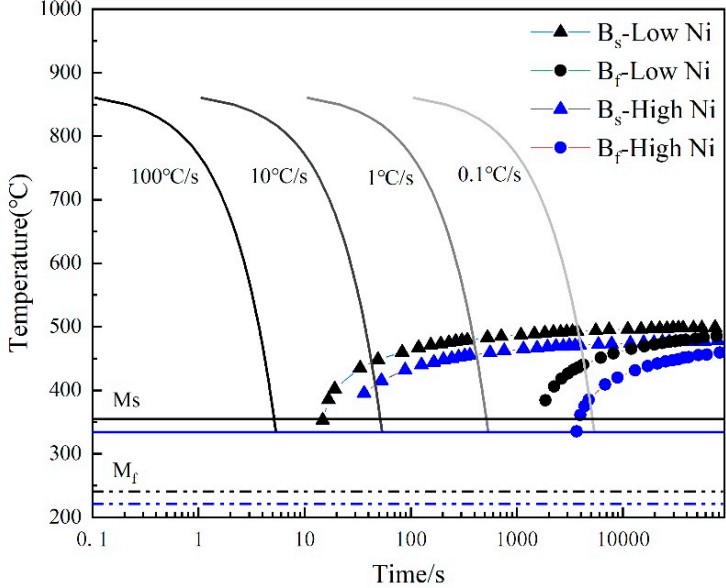

**Figure 12.** Continuous cooling transition diagram of deposited metals.

**Table 2.** Phase transition temperature of deposited metal.

| Deposited Metals | Bs/°C | Ms/°C | M$_f$/°C | ΔT(Bs-Ms)/°C |
|---|---|---|---|---|
| LN | 502.3 | 353.1 | 238.1 | 149.2 |
| HN | 481.6 | 334.9 | 218.2 | 146.7 |
| LN(26 °C/s) | 435.1 | 353.1 | 238.1 | 82.0 |
| HN(26 °C/s) | 395.1 | 334.9 | 218.2 | 60.2 |

The XRD of deposited metals (Figure 13) shows that the microstructure is α-Fe and a certain volume fraction of RA, which is calculated by (Equation (1)) is 7.80% of the LN and 3.26% of the HN. The fraction of RA decreases with the increase in Ni content. Combining bainite and martensite transformation mechanisms, the formation, volume fraction and shapes of RA are studied in detail. Firstly, the transformation of bainite is a process of carbon emission [39]. Then the carbon atoms are enriched in untransformed austenite by long-distance diffusion, which increases the stability and promotes the formation of RA. So, there are blocky and sheet RA formed during bainite transformation. Secondly, the martensite transformation is a shearing transformation. The carbon atoms exist at a supersaturated state inside martensite and nickel is especially useful in stabilizing austenite, thereby forming film RA. Meanwhile, the volume fraction of bainite reduces from 61% to 55% (Figure 11), and the volume fraction of RA decreases from 7.80% to 3.26% with the increase in Ni content. Furthermore, the RA exists as different forms of deposited metals. The RA is mainly a block and sheet of deposited metal with low Ni, while it is a thin film of that with high Ni (Figure 5). Therefore, the RA is easily produced during bainite transformation and positively correlated with the volume fraction of bainite.

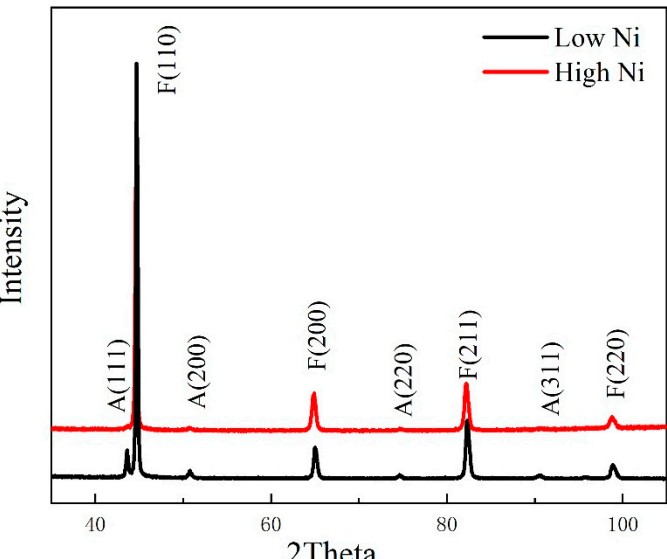

**Figure 13.** XRD patterns of deposited metals.

### 4.2. Strengthening Model of Deposited Metals

For deposited metals, the tensile strength is similar with different Ni content. It can be seen from Figure 4 that the microstructure morphology exists in the lathy structure, and the width of the lath is similar, ~500 nm. Relevant studies [40] show that when the carbon concentration is close to zero, the strength difference between martensite and bainite also approaches zero. Therefore, there is little difference between the deposited metals in terms of tensile strength. The yield strength of the HN is higher than that of the LN. However, the yield strength of martensite and bainite is not within a certain range for quantitative analysis. Therefore, the strengthening mechanism of deposited metals can be quantified through the solid solution, grain boundary, dislocation density, etc. This is discussed below.

The solid solution strengthening can be predicted according to the empirical formula (Equation (3)) [41]. The Ni and carbon elements are not included. Kennett et al. [32] believe that Ni has no substantial strengthening effect on the bainite/martensite structure, and it does not need to be considered in the solid solution strengthening. For the ferrite, carbon plays a significant role in solid solution strengthening, but for bainite/martensite tissue, the role of carbon is mainly to lead to dislocation strengthening, so it can be ignored in the solid solution strengthening.

$$\sigma_{ss} = 32\text{Mn} + 678\text{P} + 83\text{Si} + 39\text{Cu} + 11\text{Mo} - 31\text{Cr} \tag{3}$$

where $\sigma_{ss}$ is the solid solution strength, the fraction of elements is the percentage of weight.

The Hall–Patch relationship (Equation (4)) is used to calculate the grain boundary strengthening on the material.

$$\sigma_{HP} = Kd^{-1/2} \tag{4}$$

where $K$ is the Hall–Petch constant of low-carbon steel, 0.2 MPa·m$^{1/2}$ [42]; $d$ is the equivalent grain size, about 4.25 μm.

The dislocation density of deposited metals calculated by (Equation (2)) is $2.65 \times 10^{14}$ m$^{-2}$ of the LN and $3.28 \times 10^{14}$ m$^{-2}$ of the HN, respectively. The dislocation strengthening ($\sigma_{dis}$) can be estimated by the Taylor equation [43] as below:

$$\sigma_{dis} = \alpha MGb\sqrt{\rho} \tag{5}$$

where $\alpha$ is the Taylor constant, $\alpha = 0.4$; $M$ is the average Taylor factor, $M = 2.77$ [44]; $G$ is the shear modulus, $G = 80$ Gpa [45]; $\rho$ is dislocation density.

The contributions of each strengthening constituent are shown in Figure 14. The yield strength of α-Fe is about 53 MPa [46]. In this experiment, the $\sigma_{others}$ included α-Fe strength and other strengths not discussed is totally about 110 Mpa. The value of solid solution strengthening is ~80 MPa. The strengthening values of grain boundary are both ~97 MPa. The values of dislocation density strengthening are 358 MPa of the LN and 398 MPa of the HN, respectively. The dislocation density within martensite is higher than that of bainite, which is transformed at lower temperatures with high carbon content. Finally, the improvement of yield strength of the HN (41% martensite) is mainly attributed to the increase in dislocation density.

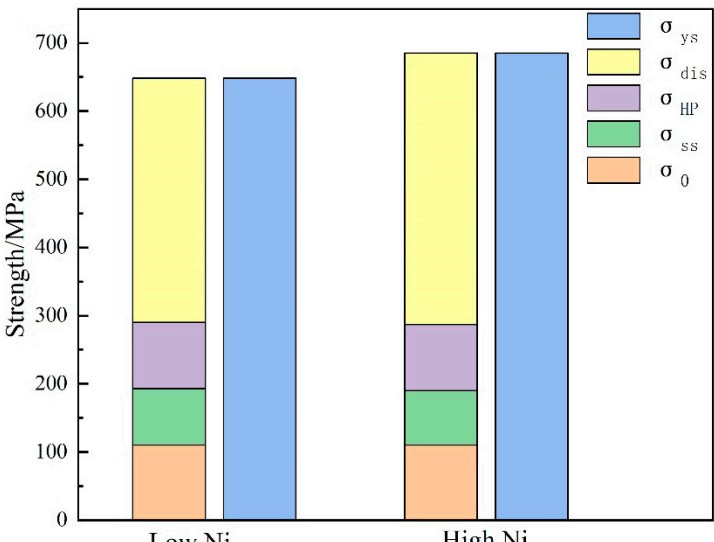

**Figure 14.** Contributions of various strengthening mechanisms to the total yield strength of deposited metals.

*4.3. Toughness of Deposited Metals*

The Charpy V-notch impact test shows that the toughness of the LN is better than the HN. It will be explained from the following three points: bainite, RA and large-angle grain boundary. Firstly, the impact energy of the LN is improved by the increase in the volume fraction of bainite (Figure 11). For multiphase microstructure, the crack propagating inside martensite will be obtuse when meeting flexible bainite. The combined deformation of bainite and martensite gives full play to the role of crack arrest and stress relief. Secondly, the volume fraction of RA is 7.80% of the LN, while it is 3.26% of the HN. Many researchers [47,48] have reported that the presence of RA is a significant role in improving toughness. During deformation, soft austenite can release internal stress and inhibit crack initiation. Then, the large misorientation angles (about 45°) between bainite/martensite and austenite boundary can consume more energy, effectively impede fracture propagation and promote toughness. Moreover, the transformation-induced plasticity effect of RA is an important mechanism for increasing toughness [49]. Finally, the area fraction of grain boundaries with large-angle ($\geq$15°) is 61.80% of the LN and 56.23% of the HN (Figure 6). The higher area fraction of misorientation angles ($\geq$15°) indicates that more grain boundaries are favorable to change the direction of crack propagation, and more energy is consumed in the process of crack propagation [50].

The new welding material is mainly considered to match 1000 MPa grade high strength low alloy steel. The B+M+RA multiphase microstructure is mainly obtained by suitable composition design. At the same time, preheating before welding is used to reduce the cooling rate of deposited metal, reduce the welding residual stress and eliminate cold crack. The stability of the arc is good in the welding process. In the process of composition design, the alloy transition coefficient has been obtained accurately and the ideal composition ratio has been acquired. At present, the high oxygen content of welding material leads to poor toughness. In order to reduce oxygen content, the improvement of the composition of welding materials is necessary, and the structure and performance of welding joints prepared by this new welding wire will be studied further.

## 5. Conclusions

This work investigates multiphase microstructure and mechanical properties of deposited metal prepared by metal-cored wire. The influence of Ni on phase transformation and the corresponding mechanical properties is discussed in detail. The following conclusions can be drawn:

- The microstructures of Ni-addition deposited metals are multiphases composed of bainite, martensite and residual austenite. The volume fraction of bainite decreases from 61% to 55%, and that of martensite increases from 37% to 41%, while the content of Ni increases from 2.5% to 4.0% because the high Ni content obviously decreases the temperature range of the bainite transformation.
- The residual austenite exists as different forms of deposited metals. The residual austenite is mainly a block and sheet of deposited metal with low Ni, while it is a thin film with high Ni. The volume fraction of residual austenite decreases from 7.8% to 3.26% with the increase in Ni content. Meanwhile, the volume fraction of residual austenite is positively correlated with that of bainite.
- The tensile strength is ~1040 MPa of deposited metals. The increase in yield strength is mainly due to the high dislocation density of deposited metals with high Ni, which have 41% martensite. The toughness of deposited metals decreases with the increase in Ni content, which is positively related to the volume fraction of bainite, residual austenite and grain boundary of large-angle.

**Author Contributions:** Designing and conducting the experiments, data analyzing, writing the paper, J.W.; supervising the experiments, X.D.; revising the paper, C.L.; assisting in the experiments, D.W. All authors have read and agreed to the published version of the manuscript.

**Funding:** This research was funded by the National Natural Science Foundation of China (grant Nos. 51804217 and 52074191), Tianjin Natural Science Foundation (grant No. 19JCQNJC02100), China Postdoctoral Science Foundation (grant No. 2019M660058) and the State Key Laboratory of Metal Material for Marine Equipment and Application (grant No. SKLMEA-K201904).

**Data Availability Statement:** The datasets used or analyzed during the current study are available from the corresponding author on reasonable request.

**Conflicts of Interest:** The authors declare no conflict of interest.

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
