# Peer review of "The Influence of Ni on Bainite/Martensite Transformation and Mechanical Properties of Deposited Metals Obtained from Metal-Cored Wire"

_metals, doi:10.3390/met11121971_

Round 1
Reviewer 1 Report
Dear Authors,
The reviewed work titled: “The influence of Ni on bainite/martensite transformation and corresponding mechanical properties of deposited metals obtained by metal cored wire” describes an experimental study aimed at evaluation of the properties of deposited metal obtained from the flux wire with a metal core (welding process: 138). The work contains elements of novelty and fits the thematic profile of the journal. Below I present the comments that should be introduced into the manuscript before deciding whether to accept the manuscript for publication.
Title: I suggest to change:
The influence of Ni on bainite/martensite transformation and mechanical properties of MCAW deposited metal
or
The influence of Ni on bainite/martensite transformation and mechanical properties of deposited metal obtained from metal cored wire
Introduction:
In my opinion, it lacks a broader description of welding with the FCAW process, which includes the MCAW process discussed in the manuscript (please add process number: 138). Of course, MCAW is also a variant of GMAW, but more often the latter abbreviation is used as a synonym for MIG/MAG processes. I think that it would be beneficial for the reader to supplement the Introduction with a paragraph (maybe second in this chapter?) concerning the characteristics of FCAW processes: classification, advantages and disadvantages of the process, types, construction and storage of additional materials, application area of individual varieties. In my opinion, it is necessary to mention the methods of wire production (seamless, copper plated) and the issues of diffusible hydrogen. In this regard, please consider relying on information from the works (I am not their co-author): https://doi.org/10.3390/ma13173888, https://doi.org/10.1016/j.jmapro.2021.06.061, 10.2478/adms-2019-0021. I think that such a supplement will be a good justification for taking up the topic of research and will have a positive effect on the future impact of the article.
In this context, I propose to replace GMAW with FCAW throughout manuscript.
Chapter 2:
The description of the preparation of research works requires more detail: how and where were the welding wires made? What was the wire diameter? Were they seamless wires? Why were 2.5 and 4% Ni wires just manufactured? What was the value of CTWD (stick out, wire extension)? What was the flow rate of shielding gas?
I suggest adding a photo of the wire cross-section and specifying filling rate (as in work (also not mine): https://doi.org/10.3390/ma13051061).
Table 1 shows the chemical composition of deposited metal. What was the method of deposition of deposited metal? Was surfacing performed on the steel plate? For example, was only the wire melted onto the copper element? The results suggest some variant of the second method (without using of base material) but this is not explained. Was deposited metal collected from the same joints used for mechanical testing samples? In Table 1, please add the chemical composition of base material and its mechanical properties.
In Figure 1, please show where the samples were collected for structural morphology investigations.
What welding machine was used?
Valid parameter writing is: "welding current" and "arc voltage".
All devices used in the research should be described in accordance with the guidelines of the journal.
Please consistently insert spaces before units.
Results:
Please enlarge all the plots, as their analysis is difficult.
In the discussion of the results, please indicate to which steel grades such wires can be used. Please also note that their use requires tests to verify their welding properties: arc stability, weld metal recovery, etc.
Acknowledgments: please either delete this section or complete it.
References are generally well chosen: please format according to journal guidelines.
Reviewer 2 Report
The authors have studied the effect of 2.5 and 4% Ni on the bainite/martensite transformation and also their affects on the mechanical properties. The manuscript is worthy for publications as the analyses are thorough and informative, however, the following questions should be addressed before the publications:
i) Are the quantities (2.5 and 4%) of Ni content optimized? The choice of Ni quantities should be explained.
ii) There are no significant changes in tensile strengths of deposited metals with Ni contents (1042±10MPa for 2.5%Ni and 1040±5MPa for 4%Ni). A detailed discussion in necessary.
iii) The proper labelling of all SEM and TEM is necessary to clear understanding of the presence of martensite and bainite.
iv) How does the misorientation angles (Fig.5) affects the mechanical properties (Fig.6) should be discussed.
v) Distributions of area fractions of misorientation angles of deposited metals for both Ni contents are same according to Fig. 9. A detail discussion is necessary to understand this figure.
vi) What could be the potential industrial application of newly designed alloys should also be mentioned in revised manuscript.
Round 2
Reviewer 1 Report
Dear Authors,
Thank you very much for the answers and introducing changes in line with my suggestions. I believe that the reviewed manuscript, after author's proofreading, can be a good source of information for readers. Additionally, lease change the vocabulary:
"powder" - "flux", "tape" - "sheath" and correct the author's name in reference [8] (it should be as in [10]).
Best regards
Author Response
Dear reviewer,
Thank you very much for your kind suggestions. The modified version is below.
Modified version
1.(line96-99) They are not the seamless wire. The dimension of the steel sheath (99.6%) outside is 12mm×0.5mm. The flux inside is metal particles. The formation process of wires included four steps: U-groove pressing formation, flux filling, rolling sealing and diameter reduction, as shown in Fig.1[28].
2.(line 411) [8]Świerczyńska, A.